# Resveratrol, Curcumin and Piperine Alter Human Glyoxalase 1 in MCF-7 Breast Cancer Cells

**DOI:** 10.3390/ijms21155244

**Published:** 2020-07-24

**Authors:** Betina Schmidt, Christian Ferreira, Carlos Luan Alves Passos, Jerson Lima Silva, Eliane Fialho

**Affiliations:** 1Laboratório de Alimentos Funcionais, Instituto de Nutrição Josué de Castro, Universidade Federal do Rio de Janeiro, Rio de Janeiro 21941-902, Brazil; schmidt_betina@yahoo.co.uk (B.S.); christianferreira_83@hotmail.com (C.F.); lu.passos@live.com (C.L.A.P.); 2Laboratório de Termodinâmica de Proteínas e Estruturas Virais Gregório Weber, Instituto de Bioquímica Médica Leopoldo de Meis, Instituto Nacional de Ciência e Tecnologia de Biologia Estrutural e Bioimagem, Universidade Federal do Rio de Janeiro, Rio de Janeiro 21941-902, Brazil; jerson@bioqmed.ufrj.br

**Keywords:** resveratrol, curcumin, piperine, GLO1, MCF-7 Cells

## Abstract

Breast cancer is the leading cause of cancer mortality in women worldwide. Conventional cancer treatment is costly and results in many side effects. Dietary bioactive compounds may be a potential source for breast cancer prevention and treatment. In this scenario, the aim of this study was to investigate the effects of the bioactive compounds resveratrol, curcumin and piperine (R-C-P) on MCF-7 breast cancer cells and to associate them to Glyoxalase 1 (GLO1) activity. The findings indicate that R-C-P exhibits cytotoxicity towards MCF-7 cells. R-C-P decreased mitochondrial membrane potential (ΔΨm) by 1.93-, 2.04- and 1.17-fold, respectively. Glutathione and N-acetylcysteine were able to reverse the cytotoxicity of the assessed bioactive compounds in MCF-7 cells. R-C-P reduced GLO1 activity by 1.36-, 1.92- and 1.31-fold, respectively. R-C-P in the presence of antimycin A led to 1.98-, 1.65- and 2.16-fold decreases in D-lactate levels after 2 h of treatment, respectively. Glyoxal and methylglyoxal presented cytotoxic effects on MCF-7 cells, with IC_50_ values of 2.8 and 2.7 mM and of 1.5 and 1.4 mM after 24 and 48 h of treatment, respectively. In conclusion, this study demonstrated that R-C-P results in cytotoxic effects in MCF-7 cells and that this outcome is associated with decreasing GLO1 activity and mitochondrial dysfunction.

## 1. Introduction

Breast cancer is the type of cancer that affects women the most worldwide, in both developing and developed countries. According to data provided by WHO, 655,690 deaths due to breast cancer in women were estimated for 2020 worldwide and is the leading cause of cancer mortality among women. Many chemotherapies capable of suppressing breast cancer growth and of preventing metastasis are available, although they are associated with side effects [1]. Breast cancer has been classified into different subtypes based on molecular and/or genetic profiles and receptor status [2,3]. MCF-7, characterized as a primary tumour positive for oestrogen and progesterone receptors, is the most studied breast cancer cell line worldwide [4].

Currently, about 50% of all drugs used in neoplasia treatment contain either natural origin active principles or semisynthetic derivatives, evidencing the importance of the search for new drugs from natural products [5]. Resveratrol (Figure 1A) is a polyphenol produced by several plants, including grapes (*Vitis vinifera* L.), red wine, berries and peanuts and has been described as displaying anti-inflammatory, antioxidant, antifungal, antibacterial, antiviral and anti-*Leishmania amazonensis* activities and effects on type 2 diabetes, cardiovascular and cognitive diseases, as well as cancer [6,7,8,9,10,11]. Resveratrol can suppress metastasis and can regulate chemoresistance, invasion and migration of cancer cells in many cancer models including breast, pancreatic, lung, skin and prostate cancer [12]. Curcumin (Figure 1B) is a polyphenolic compound in turmeric (*Curcuma longa*) rhizomes that displays potential effects against a wide variety of chronic diseases including cardiovascular, inflammatory, metabolic, neurological and skin conditions; various infectious diseases; and cancer [13,14]. Curcumin inhibits breast cancer cell proliferation by inducing cell cycle arrest and p53-dependent apoptosis, altering the expression of signalling proteins, downregulating transcription factors, and inhibiting tumour growth and angiogenesis [15]. Piperine (Figure 1C) is an active constituent of *Piper nigrum* L. and *Piper longum* and exhibits various biological properties, such as anti-inflammatory, immunomodulatory [16], antitumour [17] and anti-*Leishmania amazonensis* activities [18]. Piperine exert anti-breast cancer properties, mainly by inhibiting proliferation, migration, by promoting apoptosis, by potentiating the cytotoxicity of anticancer drugs and by reversing multidrug resistance [19].

Protein glycation is an irreversible, nonenzymatic posttranslational modification of protein amino groups (N-terminal and arginine/lysine side chains) by carbonyl compounds, leading to the formation of advanced glycation end-products (AGEs). The reaction initiates via Schiff base formation, followed by Amadori rearrangement and AGE formation, termed the Maillard reaction [20,21]. Alpha-oxoaldehydes, such as methylglyoxal (MGO) (Figure 1D), glyoxal (GO) (Figure 1E) and 3-deoxyglucosone (3-DG) (Figure 1F), are also known to form AGEs and have been associated with both diabetes and cancer [22]. AGES are generated by glucose autoxidation, lipid peroxidation and interruption of glycolysis by reactive oxygen species (ROS). Generally, the circulating AGE levels are correlated with age, oxidative stress and insulin resistance and is associated with a wide range of diseases, such as cancer [23,24]. This effect is prevented by MGO scavengers, GLO1 overexpression and N-acetyl-cysteine (NAC), a thiolic amino acid, that has been reported to scavenge free radicals, to replenish reduced glutathione (GSH), to prevent its depletion and to inhibit lipid peroxidation [25,26,27]. High glycolytic tumour rates are thought to be a hypoxia adaptation in premalignant lesions and are particularly clear in invasive metastasis tumours. One of the consequences of high glycolytic rates is an increased formation of AGEs [28].

The glyoxalase system is involved in the detoxification of reactive MGO and GO formed metabolically and enzymatically when glucose is degraded. Glyoxalase 1 (GLO1) catalyses the conversion of cytotoxic MGO to nontoxic hemithioacetal using reduced glutathione (GSH) as a cofactor. This enzyme is ubiquitously expressed in all mammalian cells and has been suggested as involved in cellular aging and death. The glyoxalase system protects cells by converting GO and MGO to D-lactate [29,30]. Bladder, breast, colon, lung, prostate and gastric tumours display increased GLO1 activity as one of the mutations processes observed during carcinogenesis [31,32].

In this context, the present study investigated the resveratrol, curcumin and piperine effects on oestrogen receptor-positive MCF-7 breast cancer cells, associating them with GLO1 activity.

## 2. Results

### 2.1. Effect of Resveratrol, Curcumin and Piperine on MCF-7 Cell Viability

MCF-7 cells were used to evaluate the effects of resveratrol, curcumin and piperine on breast cancer. First, cytotoxicity in the presence of each independent compound at different concentrations was evaluated by the 3-(4,5-dimethylthiazol-2-yl)-2,5-diphenyltetrazolium bromide (MTT) method. Resveratrol, curcumin and piperine presented a time- and dose-dependent cytotoxic effect on MCF-7 cells, with IC_50_ values of 131.0, 24.5 and 94.5 µM after 24 h of treatment and of 83.9, 11.4 and 38.3 µM after 48 h, respectively (Figure 2A–C).

Next, the capacity of each compound to damage cell membrane integrity was assessed by the trypan blue dye exclusion assay. The findings indicate that 32.12%, 60.17% and 27.41% of the cell growth were inhibited following 24 h of treatment with 131.0 µM resveratrol, 24.5 µM curcumin and 94.5 µM piperine, respectively, compared to the untreated control (Appendix A).

### 2.2. Effects of Resveratrol, Curcumin and Piperine on ΔѰm

Mitochondria play an important role in glycolytic metabolism, and in order to understand the reasons for cytotoxicity generated by compounds to MCF-7 cell line, we treated the cells with IC_50_ values of the compounds and investigated the mitochondrial membrane potential (ΔѰm) through the 5,5′,6,6′-tetrachloro 1,1′,3,3′-tetraethyl benzimidazolyl carbocyanine iodide (JC-1) assay. The treatments with 131.0 µM of resveratrol and 24.5 µM of curcumin for 24 h significantly decreased ΔѰm by 1.93- and 2.04-fold, respectively. Piperine at 94.5 µM resulted in a 1.17-fold reduction in ΔѰm (Figure 2D).

### 2.3. Protective Effects of GSH and NAC on Cell Viability

After noting that different GSH and NAC concentrations did not affect MCF-7 cell viability (Appendix A), we evaluated if pretreatment with NAC or GSH for 30 min would protect MCF-7 cells from resveratrol, curcumin and piperine cytotoxicity. Our results indicate that both GSH and NAC were able to reverse the toxic effects of all compounds in MCF-7 cells. GSH concentrations of 50, 75, 100 and 300 µM in the presence of 131.0 µM of resveratrol increased the number of viable cells by 63.06%, 69.49%, 70.79% and 79.29%, respectively (Figure 3A). In the presence of 24.5 µM of curcumin and 94.5 µM of piperine, the same GSH concentrations increased the number of viable cells by 64.54%, 73.88%, 85.68% and 76.88% and by 60.48%, 71.79%, 76.21% and 79.95%, respectively (Figure 3C,E). NAC concentrations of 0.5, 1, 2 and 5 mM in the presence of 131.0 µM resveratrol increased the number of viable cells by 70.37%, 75.82%, 87.85% and 100.54%, respectively (Figure 3B). In the presence of curcumin and piperine, the same NAC concentrations increased the number of viable cells by 61.28%, 67.47%, 71.42% and 70.05% and by 65.52%, 74.58%, 79.17% and 86.22%, respectively (Figure 3E,F).

### 2.4. GLO1 Activity

The influence of resveratrol, curcumin and piperine on GLO1 activity in MCF-7 cell extracts after 24 h of treatment was assessed. Resveratrol, curcumin and piperine reduced GLO1 activity by 1.36-, 1.92- and 1.31-fold, respectively, after 24 h of treatment (Figure 4). Piperine in the presence of GSH was able to reduce GLO1 activity by 1.24-fold (Figure 4). The inhibitory effects of resveratrol, curcumin and piperine on GLO1 activity are dose dependent (Appendix A).

The effects of resveratrol, curcumin and piperine on GLO1 enzyme expression were also assessed. MCF-7 cells were treated for 24 h with the IC_50_ values of the bioactive compounds, and Western Blot assays were performed. The results indicate no effect on GLO1 expression in MCF-7 cells treated with resveratrol and piperine compared to the control. However, curcumin decreased GLO1 expression in 1.22-fold. Thus, it seems that resveratrol and piperine influence GLO1 activity in a posttranslational manner, since they do not alter its expression (Appendix A).

### 2.5. D-Lactate Production

We evaluated the production of D-lactate in MCF-7 cells after treatment with resveratrol, curcumin and piperine in the presence or absence of antimycin A, a complex III mitochondrial electron chain inhibitor. The findings demonstrate that resveratrol, curcumin and piperine were not able to decrease D-lactate production. However, resveratrol and curcumin in the presence of antimycin A decreased D-lactate levels by 1.84- and 1.98-fold and by 1.55- and 1.65-fold after 60 and 120 min, respectively (Figure 5A,B), while piperine inhibited D-lactate levels by 2.01-, 1.85- and 2.16-fold, respectively, after 45, 60 and 120 min of treatment (Figure 5C).

### 2.6. AGEs Toxicity

In order to determine the effects of MGO, GO and 3-DG on MCF-7 cell viability, cells were treated with different α-oxaldehydes concentrations for 24 and 48 h and the percentage of viable cells were defined by the MTT assay. GO and MGO presented a cytotoxic effect on MCF-7 cells, with IC_50_ values of 2.8 and 2.7 mM after 24 h of treatment and of 1.5 and 1.4 mM after 48 h, respectively (Figure 6A,B). Unlike the other two metabolites, 3-DG did not cause toxic effects at the tested concentrations at 24 and 48 h (Figure 6C).

Next, we determined the effects of MGO, GO and 3-DG in the presence of resveratrol, piperine and curcumin on MCF-7 cell viability. Our results demonstrate that resveratrol at 131.0 µM and curcumin at 24.5 µM associated to 2.7 mM MGO were able to kill MCF-7 cells, reducing the percentage of viable cells after 24 h of treatment by 1.83 and 2.14-fold in relation to MGO (Table 1). Piperine also decreased MGO- and GO-treated cell viability by 1.96 and 1.63 after 24 h and by 2.56- and 1.66-fold after 48 h, respectively (Table 1).

## 3. Discussion

Functional foods are composed of biologically active substances that can stimulate physiological or metabolic processes, thus reducing the risk of disease and maintaining health. However, in order to provide positive effects, functional foods should be part of daily diets so that active compounds are constantly present in the body [33]. Our results indicate that resveratrol, curcumin and piperine cause cytotoxic effects on MCF-7 cells. This result agrees with previous findings reporting resveratrol, curcumin and piperine inhibition of breast cancer cell proliferation in vitro. We have previously reported that resveratrol in association with melphalan triggered cytotoxic effects in MCF-7 and MDA-MB-231 cells in vitro [7] and increased fatty acids in MCF-7 and MDA-MB-231 cells [34]. Wang et al. [35] demonstrated that resveratrol inhibits the growth of MDA-MB-231 cells in a dose-dependent manner as well as their solid lipid nanoparticle form, displaying IC_50_ values of 72.06 and 40.82 µg/mL, respectively. Ediriweera et al. [36] reported that curcumin has an effect on MCF-7 cells, with an IC_50_ of 3.5 µM after 48 h of incubation. In addition, IC_50_ values in MCF-7 and SKBR3 cells for piperine are approximately 200 μM and 50 μM, respectively, after 48 h of incubation [37].

Since the discovery that alterations in the inner mitochondrial membrane represent a critical step in the regulation of intrinsic apoptotic pathway, mitochondria have been considered an important organelle concerning proliferation control and cell death as well regarding various cell survival aspects. Increased reactive oxygen species (ROS) levels may trigger decreased ΔѰm [38] which may, in turn, lead to cell death. Kocyigit and Guler [39] demonstrated that curcumin decreases cell viability and ΔѰm while increasing DNA damage, apoptosis and ROS levels in both melanoma cancer and normal cells in a dose-dependent manner and that these activities were significantly higher in melanoma cells than in normal cells at higher concentrations. Mouse colon carcinoma CT26 cells treated with 20.0 µM resveratrol or 40.0 µM piperine and mouse melanoma B16F10 cells treated with 10.0 µM resveratrol or 30.0 µM piperine exhibited significantly augmented ionizing radiation-induced apoptosis and loss of ΔѰm through enhanced ROS generation [40].

The decreased mitochondrial membrane potential may be altered by increasing ROS levels or mitochondrial protein glycation [41,42]. Because of this, we investigated whether GSH and NAC antioxidant exposure could result in protective effects on the viability of treated cells. Our findings indicate that both GSH and NAC were able to reverse the cytotoxicity of bioactive compounds on MCF-7 cells. NAC is a nonenzymatic antioxidant that acts primarily by blocking the lipid peroxidation chain, by eliminating oxygen or by chelating metal ions. It also promotes the synthesis of GSH, increasing cysteine availability, thereby raising gamma-glutamyl cysteine synthetase levels, an enzyme that limits the GSH formation route [43]. Therefore, NAC may generate the substrate for GSH formation, thus optimizing GLO1 activity on the detoxification of α-oxaldehydes.

Most cancer cells perform aerobic glycolysis to generate lactate, commonly known as the Warburg effect and accompanied by increased formation of the cytotoxic metabolite MGO. Glyoxalase I belongs to the glyoxalase enzyme system that converts MG to D-lactate. The GLO1 substrate is hemithioacetal, formed nonenzymatically from the nucleophilic reaction between the cellular tripeptide GSH and MGO [29]. It is acknowledged that the glyoxalase enzyme system is involved in the regulation of tumour cell metabolism, proliferation, migration and survival [44]. Because of this, an urgent need for the development of GLO1 inhibitor drugs for clinical cancer therapy is noted. The present study demonstrates that resveratrol, curcumin and piperine dose-dependently reduce GLO1 activity while only curcumin reduces GLO1 expression in MCF-7 cells extracted after 24 h of treatment. Santel and colleagues [45] found that specific GLO1 activity in curcumin-treated JIMT-1 breast carcinoma cells was lower than in non-treated cells, decreasing dose-dependently. In addition, Meiyanto et al. [46] reported that curcumin finely binds to GLO1 and induces autophagy in 4T1 cells.

We evaluated GLO1 activity in MCF-7 cells treated with each isolated bioactive compound (resveratrol, curcumin or piperine) for 24 h and compared the results to GLO1 activity in cells pretreated with GSH and after treatment with the isolated compounds. The results indicate a relationship between the cytotoxic effect of the bioactive compounds on MCF-7 cells and decreased GLO1 activity in a dose-dependent way. The same protective effect of GSH on bioactive compound action towards GLO1 activity was observed in cells pretreated with GSH, an antioxidant and GLO1 cofactor, which was also observed concerning the cytotoxicity of bioactive compounds in MCF-7 cells. Other bioactive compounds inhibit GLO1. Yamamoto et al. (2019) demonstrated that passion fruit seed extracts and stilbenes piceatannol and scirpusin B inhibited GLO I activity in NCI-H522 human cancer cells that express high levels of GLO I [47]. These data suggest that the possibility of protection generated by antioxidant pretreatment is related to the optimization of the α-oxaldehydes detoxification pathway by the glyoxalase system. 

D-lactate is the end-product of the GLO2 enzymatic reaction produced in the MGO pathway, and glyoxalase inhibition is known to result in decreased D-lactate release. In this sense, we demonstrated that resveratrol, curcumin and piperine did not decrease D-lactate production. In contrast, it has been reported that 20 µM curcumin in 1321N1 brain astrocytoma cells decreases D-lactate release after 24 h of treatment [45]. Piperine induces intracellular lactate release to activate the AMP-activated protein kinase (AMPK) signalling pathway involved in mitochondrial respiration in C2C12 cells [48]. Resveratrol has been noted as reducing the glucose flux and lactate production in ovarian cancer cells [49].

We have demonstrated that resveratrol, curcumin and piperine inhibited D-lactate in the presence of antimycin A. This confirms the ΔѰm variations, indicating that resveratrol, curcumin and piperine generate damage to MCF-7 cell mitochondria.

α-dicarbonyl compounds, including MGO, GO and 3-DG, are highly reactive concerning the amino group of proteins in order to form AGEs [50]. We demonstrated that GO and MGO caused cytotoxic effects in MCF-7 cells while 3-DG did not shift after 24 and 48 h of treatment (Figure 6). MG and GO can be produced by the autoxidation of carbohydrates and lipid peroxidation in most glucose-metabolizing cells. 3-DG, which can be formed by the non-oxidative rearrangement and hydrolysis of Amadori products, reacts rapidly with protein amino groups to form AGEs such as imidazolone, pyrraline and carboxymethyllysine [23,51]. It is described that MG and GO are potentially more reactive than 3–DG [52]. The lower reactivity potential of 3-DG compared to MG and GO and its synthesis pathway is likely to justify the different results obtained.

Reactive carbonyl species like MGO and GO react with proteins, phospholipids and DNA and produce advanced glycation end-products (AGEs) presenting pro-oxidant and pro-inflmammatory properties but, in some cases, when associated with certain antioxidant compound, such as NAC, are able to reverse these properties, as reported in colorectal cancer [53]. Shen et al. [54] demonstrated resveratrol present anti-glycation activity in a dose-dependent manner in the cell free system of BSA-fructose, BSA-MGO and arginine-MGO due to a conjugation reaction. Sun et al. [55] suggested that curcumin inhibits the formation of AGEs through its MGO-trapping ability in endothelial cells.

Chronic stress or food consumption situations lead to MGO-related cell, protein and DNA damages and may be related to cancer development processes [56]. However, in cancer cells exhibiting normally high glucose demands, exacerbated glucose consumption generates ROSs which, in turn, increase Poly (ADP-ribose) polymerase (PARP) and inhibit Glyceraldehyde-3-phosphate dehydrogenase (GAPDH) by increasing the amount of glyceraldehyde-6-phosphate and by creating a pathway for the MGO formation [57]. Thus, the production of toxic MGO is increased in cancer cells, and as a form of defence, increased GLO1 expression and glyoxalase system optimization would be noted among the mutagenic aspects that cancer cells suffer [30,58]. In the present study, we demonstrated that MGO and GO are toxic to MCF-7 cells, emphasizing the importance of GLO1 activity not only for the survival of this cell type but also as a target in anticancer therapy.

## 4. Materials and Methods

### 4.1. Reagents

Resveratrol, curcumin, piperine and 3-(4,5-dimethylthiazol-2-yl)-2,5-diphenyltetrazolium bromide (MTT) were obtained from Sigma-Aldrich (St. Louis, MO, USA). Dulbecco’s minimal essential medium (DMEM) and foetal bovine serum (FBS) used in the cell culture methods were purchased from Gibco (Grand Island, NY, USA). All other chemicals were purchased in the purest commercially available form.

### 4.2. Cell Culture

The human breast epithelial cell line MCF-7 was obtained from American Type Culture Collection (ATCC) and grown in DMEM supplemented with 10% FBS, 100 units/mL of penicillin, 100 mg/mL of streptomycin and 5 mg/mL of insulin. The cells were kept at 37 °C in a humidified 5% CO_2_ atmosphere according to ATCC recommendations. The cells were counted and plated at the same initial density in all experiments. After reaching 70–80% confluence, the cells were treated with various concentrations of the assessed compounds at different times.

### 4.3. Cell Viability Assay

The cell viability assay was performed using the MTT method. After treatment with 50, 100, 200, 300, 400 and 500 µM of resveratrol; 1, 5, 10, 25, 50, 75 and 100 µM of curcumin; and 1, 5, 10, 25, 50, 75, 100, 150, 200 and 300 µM of piperine for 24 or 48 h, the cells were washed with phosphate buffered saline (PBS) and incubated for 3 h in 0.5 mL of an MTT solution (0.5 mg/mL of PBS) at 37 °C under a 5% CO_2_ atmosphere in an incubator. Subsequently, the medium was removed, and 0.5 mL of absolute isopropanol was added to the attached cells. The absorbance of the converted dye in living cells was determined at 595 nm. The cell viability of MCF-7 cells cultured in the presence of the assessed compounds was calculated as a percent of the control cells, and the IC_50_ values were obtained from dose-response curves. All experiments were performed in triplicate, and the GraphPad Software 6.0 (GraphPad Inc., San Diego, CA, USA) was used for the IC_50_ calculation.

### 4.4. ΔΨm Determinations

Changes in MCF-7 ΔΨm (mitochondrial membrane potential) were determined using a mitochondrial staining kit (Sigma-Aldrich, St. Louis, MO, USA). This dye accumulates in the mitochondrial matrix under the influence of ΔΨm, and its monomeric form increases in unhealthy or apoptotic cells. The cells were treated with or without IC_50_ concentrations determined for resveratrol, curcumin and piperine for 24 h. A JC-1 (10 µg/mL) staining solution prepared according to the manufacturer’s instructions was added to 10^6^ cells for 10 min at 37 °C. The ΔΨm was determined at 490 nm excitation and 530 nm emission wavelengths for JC-1 monomers and at 525 nm excitation and 590 nm emission wavelengths for J-aggregates using a fluorimeter equipped with 96-well opaque culture plates. The values were expressed by the emission ratio between red/green fluorescence (590/530).

### 4.5. D-Lactate Levels

Lactate determinations were performed in a hydrazine/glycine buffer (pH 9.2) containing 5 mg/mL β-NAD^+^ and 15 units/mL lactate dehydrogenase in a final volume of 200 μL. Absorbances due to NADH formation were monitored using a spectrophotometer for 30 min at 340 nm and were correlated to the presence of lactate in the samples by interpolation with a standard curve [59].

### 4.6. Protein Determinations

Protein sample concentrations were determined as described by Lowry et al. [60].

### 4.7. Glyoxalase I Activity Assay

Glyoxalase I activity was determined according to Hansen et al. [61]. The assay was carried out in 96-well microplates using a microplate spectrophotometer (UV Star—Greiner). The reaction mixture (200 µL/well) contained 50 mM sodium-phosphate buffer pH 7.2, 2 mM methylglyoxal (MG) and 1 mM glutathione (GSH) (preincubated for 30 min at room temperature). A total of 10–15 µg of protein for each sample per well was added to the buffer. The formation of S-(D)-lactoylglutathione was linear and monitored at 240 nm for 15 min at 25 °C. One glyoxalase I activity unit is defined as the amount of this enzyme that catalyses the formation of 1 µmol of S-(D)-lactoylglutathione per minute. Specific activity is expressed as milliunits per milligram of protein.

### 4.8. Western Blot

For protein extract preparation after treatment with or without the IC_50_ values determined for resveratrol, curcumin and piperine for 24 h, cells were washed with PBS and lysed in liquid nitrogen. Cells were then scraped using a lysis buffer (10 mM ethylenediamine tetraacetic acid, 5 mM Tris-HCl, 5 mM sodium fluoride, 1 mM phenylarsine oxide, 1 mM sodium orthovanadate, 1 µM okadaic acid and 1 mM phenylmethylsulfonyl fluoride; pH 7.4) containing a protease inhibitor cocktail (Sigma-Aldrich, St. Louis, MO, USA). The lysates were then collected, sonicated and cleared by centrifugation for 5 min at 4 °C, and supernatants were stored at −80 °C. Equal amounts of total cellular proteins (100 µg) were applied to sodium dodecyl sulphate-polyacrylamide gels (SDS-PAGE) and transferred onto polyvinylidene difluoride (PVDF) membranes (Immobilon P, Millipore, MA, USA). Membranes were then blocked for 2 h in a tris-buffered saline containing 1% Tween 20 (TBS-T) and 5% non-fat milk and incubated for 2 h with the anti-GLO1 primary antibody (1:1000, Sigma-Aldrich, St. Louis, MO, USA), washed with TBS-T and finally incubated with a peroxidase-conjugated secondary antibody (1:5000, Sigma-Aldrich, St. Louis, MO, USA) for 2 h. Protein bands were visualized with the enhanced chemiluminescence (ECL) kit (Amersham, UK) using a C-DiGit Chemiluminescent Western Blot Scanner (LI-COR Biotechnology, Lincoln, Nebraska, USA). Images were analysed using the Image J 1.50i software (NIH, USA), and results were expressed as arbitrary units, calculated by the fraction of pixels presented by each band relative to the GAPDH bands.

### 4.9. Statistical Analysis

The data were analysed by the Shapiro–Wilk normality test, followed by Student’s *t*-test when comparing two groups and by a one-way ANOVA for more than two groups using the GraphPad Prism 6 software. *p* values equal or less than 0.05 were considered significant.

## 5. Conclusions

In conclusion, this study demonstrated that resveratrol, curcumin and piperine exposure results in cytotoxic effects in MCF-7 cells, associated with decreasing GLO1 activity and mitochondrial cell damage. Other aspects involving the action of these compounds in glyoxalase system enzyme expression as well as the mRNA expression of glyoxalases 1 and 2 should be further investigated as chemotherapeutic targets.

## Figures and Tables

**Figure 1 ijms-21-05244-f001:**
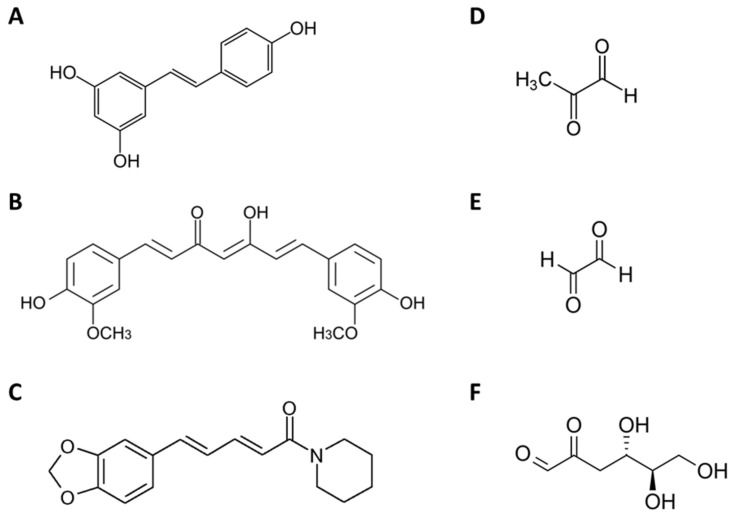
Chemical structure of resveratrol (**A**), curcumin (**B**), piperine (**C**), methylglyoxal (**D**), glyoxal (**E**) and 3-deoxyglucosone (**F**).

**Figure 2 ijms-21-05244-f002:**
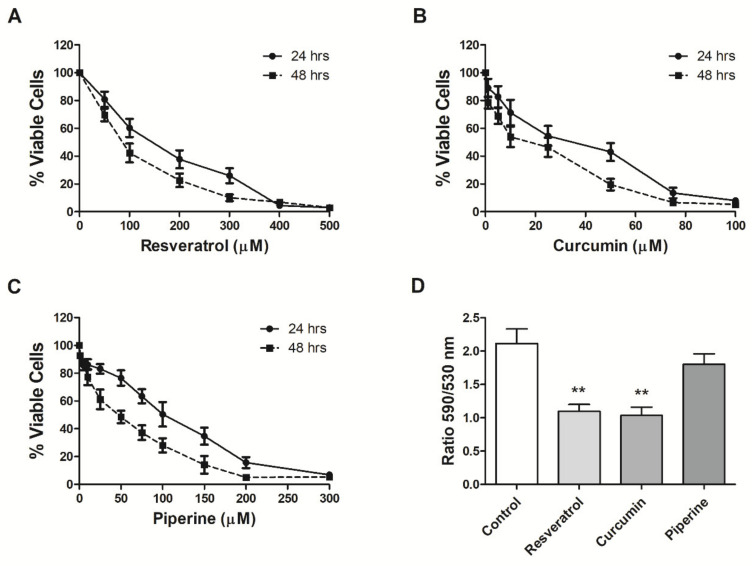
Resveratrol, curcumin and piperine MCF-7 cell viability and mitochondrial membrane potential: Cells were treated for 24 (●) and 48 (■) h with different resveratrol (**A**), curcumin (**B**) or piperine (**C**) concentrations, and cell viability was then determined by the 3-(4,5-dimethylthiazol-2-yl)-2,5-diphenyltetrazolium bromide (MTT) assay. (**D**) Cells were treated for 24 h with 131.0 µM resveratrol, 24.5 µM curcumin or 94.5 µM piperine, and mitochondrial membrane potential was determined by 5,5′,6,6′-tetrachloro 1,1′,3,3′-tetraethyl benzimidazolyl carbocyanine iodide (JC-1) staining. The results are representative of three experiments performed in triplicate ± SEM. ** *p* < 0.001 in relation to the control.

**Figure 3 ijms-21-05244-f003:**
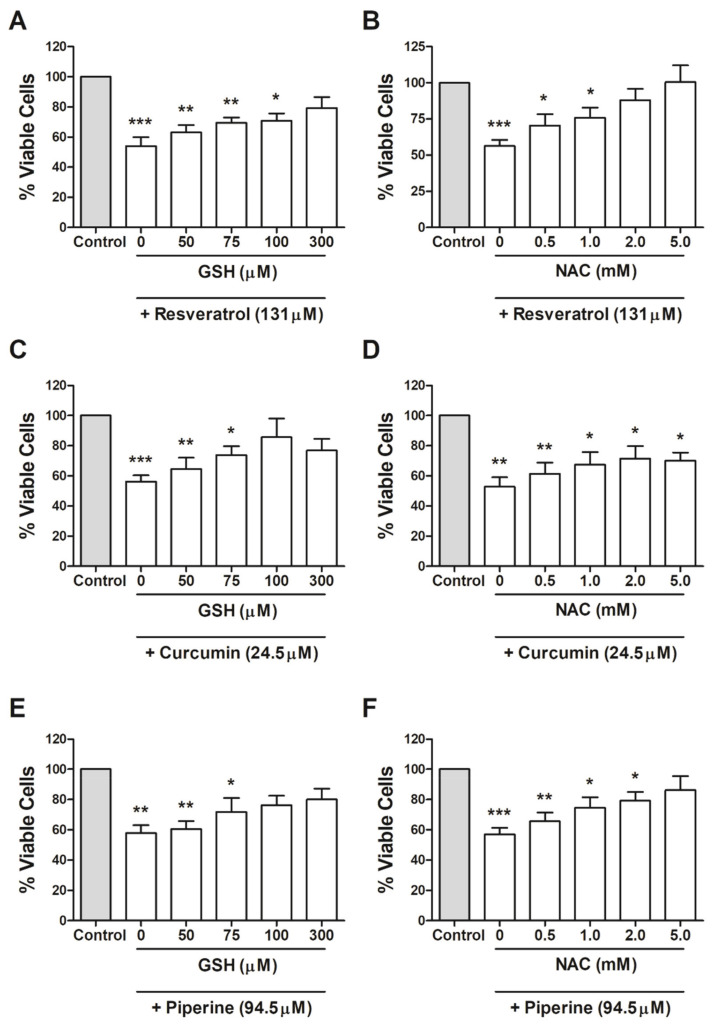
Protective effect of glutathione (GSH) and N-acetyl-cysteine (NAC) on MCF-7 cell viability after resveratrol, curcumin and piperine treatment: Cells were treated with 50, 75, 100 and 300 µM of GSH or 0.5, 1.0, 2.0 and 5.0 mM of NAC, and 131.0 µM resveratrol (**A**,**B**), 24.5 µM curcumin (**C**,**D**) or 94.5 µM piperine (**E**,**F**) were added after 30 min,. After 24 h, cell viability was determined by the MTT assay. The results are representative of three experiments performed in triplicate ± SEM. * *p* < 0.05, ** *p* < 0.001, and *** *p* < 0.0001 in relation to the control.

**Figure 4 ijms-21-05244-f004:**
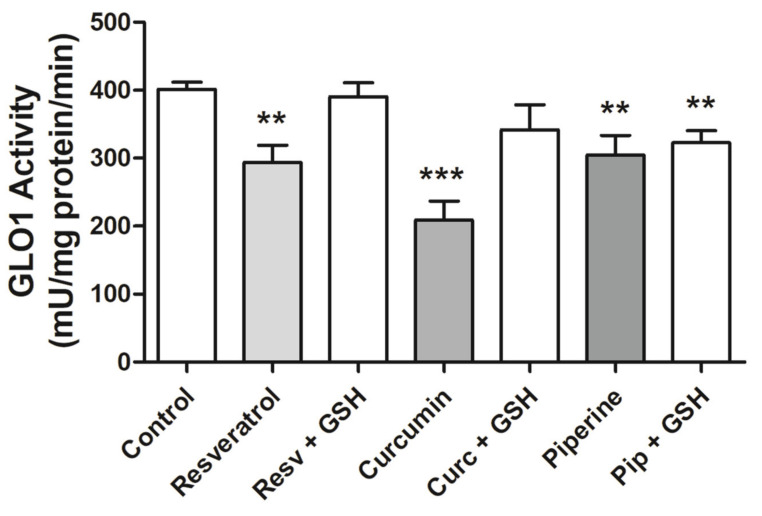
GLO1 activity in MCF-7 cells after resveratrol, curcumin and piperine treatment: Cells were treated with 100 µM of GSH, and 131.0 µM resveratrol, 24.5 µM curcumin or 94.5 µM piperine were added after 30 min. After 24 h, GLO1 activity was determined in lysed cell extracts. The results are representative of three experiments performed in triplicate ± SEM. ** *p* < 0.001 and *** *p* < 0.0001 in relation to the control. Resv = Resveratrol, Curc = Curcumin, PIP = Piperine.

**Figure 5 ijms-21-05244-f005:**
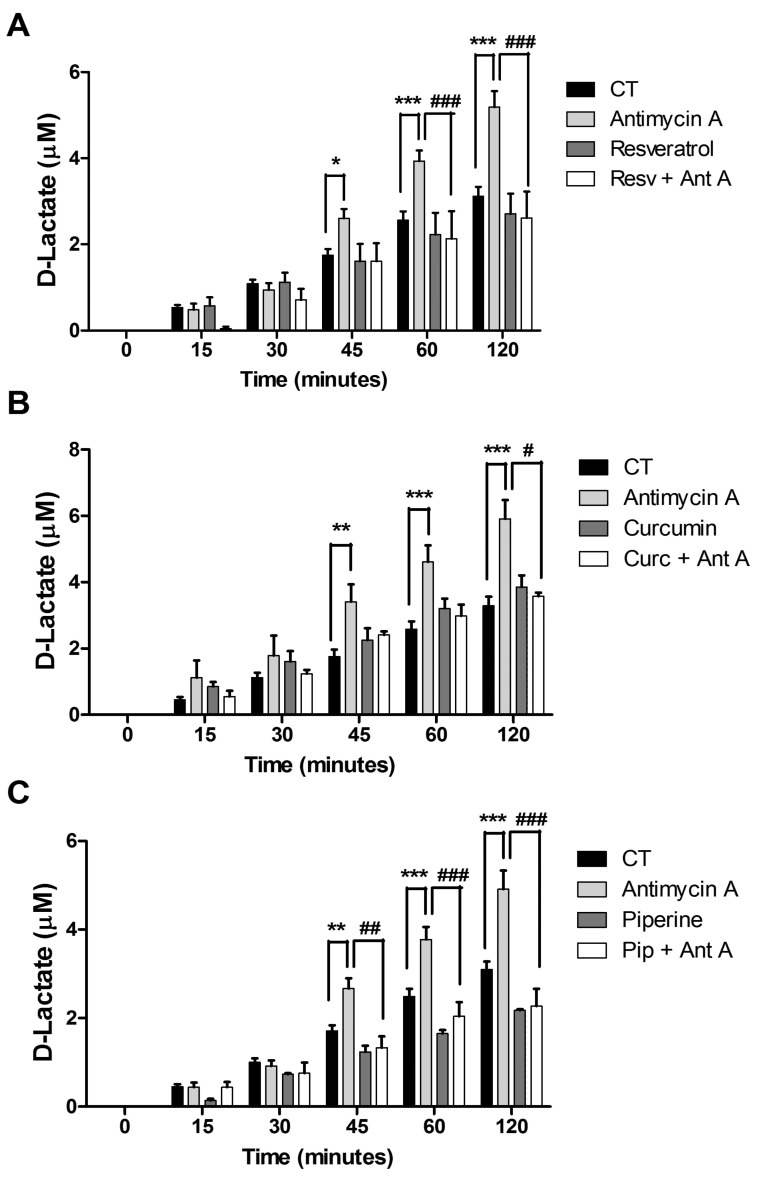
D-Lactate production in MCF-7 cells after resveratrol, curcumin and piperine treatment: Cells were treated with 131.0 µM resveratrol (**A**), 24.5 µM curcumin (**B**) or 94.5 µM piperine (**C**) for 24 h. After treatment, medium aliquots were collected at 0, 15, 30, 45, 60 and 120 min and incubated in a hydrazin buffer, pH 9.2, with NAD^+^ and lactate dehydrogenase (LDH) for released lactate measurement. Antimycin A at 2 µg/mL was used as a positive control. The results are representative of three experiments performed in triplicate ± SEM. * *p* < 0.05, ** *p* < 0.001, and *** *p* < 0.0001 in relation tothe control. ^#^
*p* < 0.05, ^##^
*p* < 0.001, and ^###^
*p* < 0.0001 in relation to antimycin. Resv = Resveratrol, Curc = Curcumin, PIP = Piperine, Ant A= Antimycin A.

**Figure 6 ijms-21-05244-f006:**
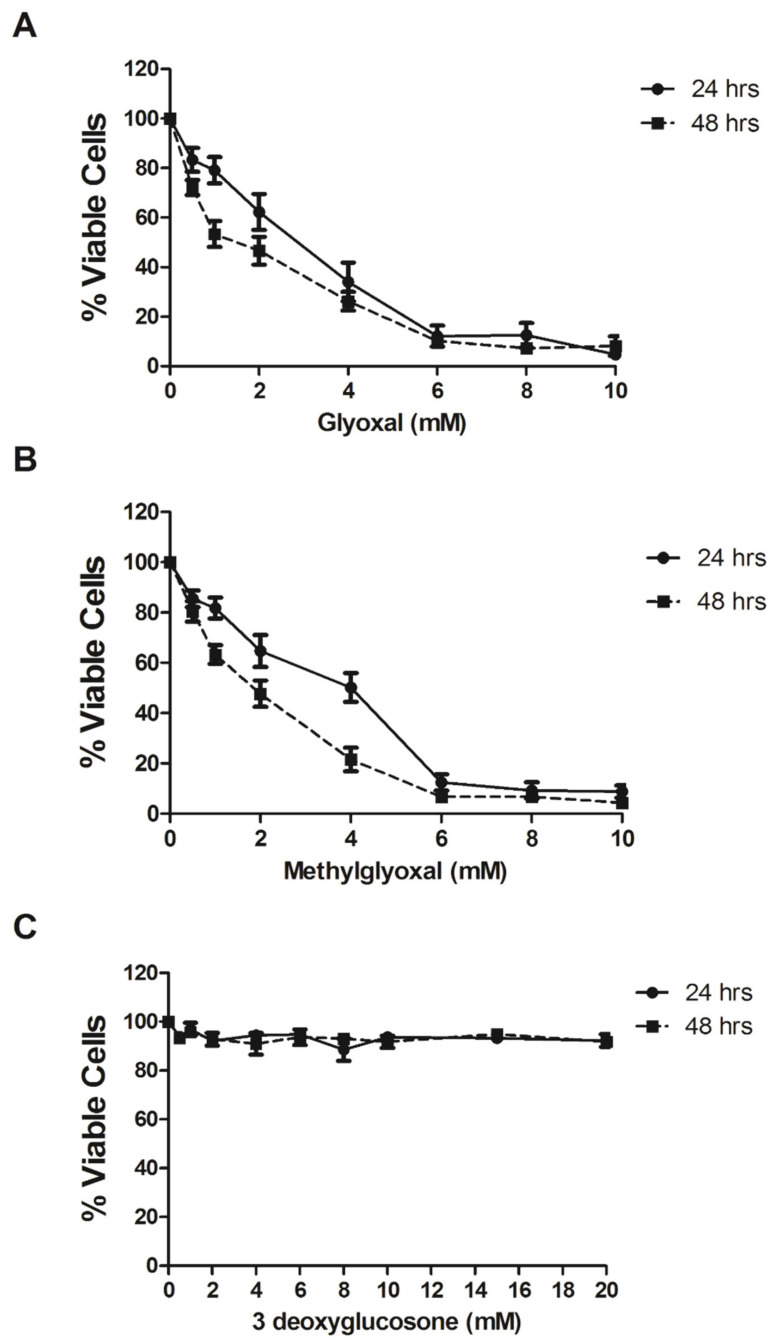
Effects of glyoxal, methylglyoxal and 3-deoxyglucosone on MCF-7 cell viability: Cells were treated for 24 (●) and 48 (■) h with different glyoxal (**A**), methylglyoxal (**B**) or 3-deoxyglucosone (**C**) concentrations, and cell viability was determined by the MTT method. The results are representative of three experiments performed in triplicate ± SEM.

**Table 1 ijms-21-05244-t001:** Effects of resveratrol, curcumin and piperine associated with methylglyoxal and glyoxal on MCF-7 cell viability.

Compounds	Percentage of Viable Cells^#^	
	24 h	48 h
2.7 mM MGO	57.10	37.86
2.7mM MGO + 131.0 µM Resveratrol	31.21	16.52
2.7 mM MGO + 24.5 µM Curcumin	26.71	14.43
2.7 mM MGO + 94.5 µM Piperine	29.16	14.79
2.8 mM GO	58.26	42.45
2.8 mM GO + 131.0 µM Resveratrol	33.38	20.45
2.8 mM GO + 24.5 µM Curcumin	35.83	18.79
2.8 mM GO + 94.5 µM Piperine	35.72	25.52

^#^The percentage of viable cells was determined by the MTT assay in relation to control cells.

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
