# Peer review of "Resveratrol, Curcumin and Piperine Alter Human Glyoxalase 1 in MCF-7 Breast Cancer Cells"

_ijms, 2020, doi:10.3390/ijms21155244_

Round 1

Reviewer 1 Report

Abstract:the full name of GLO1, GSH, NAC and ΔΨm should be given.

Introduction: why did authors choose resveratrol, curcumin and piperine? More information of

 them on the effect about breast cancer should be provided. What’s role of NAC during glycation or carcinogenesis? More detail should be mentioned in the introduction.

Line 246. “3-DG did not shafter 24 and 48 hours of treatment”,more discussion should be added.

Reference: more new research papers should be cited.

Reviewer 2 Report

In the present manuscript, Schmidt et al. found that the bioactive compounds such as resveratrol, curcumin and piperine (R-C-P) have cytotoxic effects in MCF-7 cells, and these results are related to decreased GLO1 activity and mitochondrial dysfunction. This is an interesting study and a well conducted one. However, there are several questions are still needed to be clarified.

  1. Authors should use at least two different ER+ and PR+ breast cancer cell lines in this study.
  2. In this study, authors found that the cytotoxicity of R-C-P on MCF-7 cells is associated to the decrease of GLO1 activity and mitochondrial dysfunction. However, several previous studies have indicated the similar results.
  3. Have authors do the animal experiments to confirm their in vitro results?

Author Response

REFEREE# 2

Question: Authors should use at least two different ER+ and PR+ breast cancer cell lines in this study.

Answer: Unfortunately, at this moment, due to the COVID-19 pandemic, for reasons of preservation of collective health, our institution remains closed and we don't have a date for returning to the laboratory, probably the return to normal activities will be in October 2020. Therefore, we are unable to carry out experiments, but it`s an excellent ideia for the future works.

Our research group works with MCF-7 cells (ER+ and PR+) and MDA-MB-231 cells (ER- and PR-), besides MCF-10A (non-tumorigenic cells). We showed previously that resveratrol induce a significant decrease (~40-50%) in cell viability in MCF-7 cells, while the same concentration of resveratrol causes only a minor decrease (~10-40%) in cell viability in MDA-MB-231 cells (Gomes et al., Scientific Reports, 9:13960, 2019). Additionally, we had already showed previously that IC50 values of resveratrol in association with melphalan (chemotherapeutic agent) was more effective in MCF-7 cells than MDA-MB-231 (Casanova et al., J Cell Biochem., 8:2586-96, 2012). Because resveratrol effects are more pronounced in MCF-7 cells, we decided to study the association of this compound with others (piperine and curcumin) in regulating Glyoxalase 1 (GLO1) activity and mitochondrial dysfunction in this cell line. MCF-7 cells have the following characteristics: primary tumor, presence of estrogen (ER+) and progesteron receptos (PR+) and is the most studied human breast cancer cell line in the world. The results from this cell line have had an important impact upon mammary cancer research and patient outcomes (Lee et al. MCF-7 Cells – Changing the Course of Breast Research and Care for 45 Years. Journal of the National Cancer Institute, Volume 107, Issue 7, July 2015, djv073, https://doi.org/10.1093/jnci/djv073).

Additionally, other research groups published with the effect of one bioactive compound or the association in one tumor cell line: I can cite some examples: - Ying et al. Piperine inhibits LPS induced expression of inflammatory mediators in RAW 264.7 cells. Cell Immunol. 2013; 1-2:49-54; - Yang J, Liu RH. Synergistic effect of apple extracts and quercetin 3-beta-d-glucoside combination on antiproliferative activity in MCF-7 human breast cancer cells in vitro. J Agric Food Chem. 2009; 18:8581-6; - Pal MK, Jaiswar SP, Srivastav AK, Goyal S, Dwivedi A, Verma A, et al. Synergistic effect of piperine and paclitaxel on cell fate via cyt-c, Bax/Bcl-2-caspase-3 pathway in ovarian adenocarcinomas SKOV-3 cells. Eur J Pharmacol. 2016; 791:751-762; and - Zhu GH, Dai HP, Shen Q, Ji O, Zhang Q, Zhai YL. Curcumin induces apoptosis and suppresses invasion through MAPK and MMP signaling in human monocytic leukemia SHI-1 cells. Pharm Biol. 2016; 8:1303-11.

Based on these arguments, we think that this paper will give a good return for a science community because is the fisrt work that associate three bioactive compounds that can be able to decrease GLO1 activity and mitochondrial dysfunction in breast cancer cells.

Question: In this study, authors found that the cytotoxicity of R-C-P on MCF-7 cells is associated to the decrease of GLO1 activity and mitochondrial dysfunction. However, several previous studies have indicated the similar results.

Answer: GLO1 can be a target for human tumors and the discovery of compounds that regulate the expression/activity of this enzyme is a challenge. As said before, is the first work that associates resveratrol, curcumin and piperine with the modulation of GLO1. Other bioactive compounds like anthocyanidins, mainly delphinidin, have potent inhibitory effect on human GLO1 (Takasawa et al., 2010). The same research group showed that hydroxy groups at the B ring of flavonoids present an effective inhibition of the human GLO1 (Takasawa et al., 2008). Curcumin, used in this study, can inhibit GLO1 with non-tolerable levels of MGO and GSH, which, in turn, modulate various metabolic pathways including depletion of cellular ATP and GSH contents. This may account for curcumin's potency as an anti-inflammatory and anti-tumor agent (Santel et al., 2008).

Takasawa, R.; Saeki, K.; Tao, A.; Yoshimori, A.; Uchiro, H.; Fujiwara, M.; Tanuma, S. Delphinidin, a dietary anthocyanidin in berry fruits, inhibits human glyoxalase I. Bioorg. Med. Chem. 2010. 18(19), 7029-33. doi: 10.1016/j.bmc.2010.08.012.

Takasawa, R.; Takahashi, S.; Saeki, K.; Sunaga, S.; Yoshimori, A.; Tanuma, S. Structure-activity relationship of human GLO I inhibitory natural flavonoids and their growth inhibitory effects. Bioorg. Med. Chem2008. 16(7), 3969-75. doi: 10.1016/j.bmc.2008.01.031.

Santel, T.; Pflug, G.; Hemdan, N.Y.; Schafer, A.; Hollenbach, M.; Buchold, M.; Hintersdorf, A.; Lindner, I.; Otto, A.; Bigl, M. et al. Curcumin inhibits glyoxalase 1: A possible link to its anti-inflammatory and anti-tumor activity. Plos One2008. 3 (10): e3508. doi: 10.1371/journal.pone.0003508.

Question: Have authors do the animal experiments to confirm their in vitro results?

Answer: In our laboratory, we used the experimental model of murine breast carcinoma with 4T1 cells, which is a relevant parameter model for the study of the biological and pharmacological effects of bioactive compounds against the primary site, metastasis and tumor microenvironment. However, due to the COVID-19 pandemic, our institution remains closed and we don't have a date for returning to the laboratory, probably the return to normal activities in the bioterium will be in 2021. Therefore, we are unable to carry out in vivo tests.

We believe that this work is suitable for publication in International Journal of Molecular Sciences. Since these compounds have been pointed out as putative cancer therapy agents, the results obtained with the association of bioactive compounds in cell culture indicate the potential for clinical research to support the use of adjuvant agents for breast cancer therapy.

We hope that the modifications made in the new version of the manuscript have properly addressed the criticism and suggestions made by the referees, and that the improvements made in the manuscript will be enough for its publication in International Journal of Molecular Sciences.

Best regards,

Prof. Eliane Fialho

[email protected]

Round 2

Reviewer 2 Report

I think 4T-1 (TNBC cells) is not suitable for your in vivo experiments in the future. You should use MCf-7 or other ER+/PR+ cell lines to conduct your in vivo experiments.